# Personality Traits Modulate the Effect of tDCS on Reading Speed of Social Sentences

**DOI:** 10.3390/brainsci11111464

**Published:** 2021-11-05

**Authors:** Cristian Reyes, Iván Padrón, Sara Nila Yagual, Hipólito Marrero

**Affiliations:** 1Experimental Psychology Lab, Department of Psychology, Carl von Ossietzky University of Oldenburg, 26129 Oldenburg, Germany; 2Instituto Universitario de Neurociencia, Universidad de La Laguna, 38200 San Cristóbal de La Laguna, Spain; ivpadron@ull.edu.es (I.P.); hmarrero@ull.es (H.M.); 3Departamento de Psicología Evolutiva y de la Educación, Universidad de La Laguna, 38200 San Cristóbal de La Laguna, Spain; 4Facultad de Ciencias Sociales y de la Salud, Universidad Estatal Península de Santa Elena, La Libertad 241702, Ecuador; pssarukita@hotmail.com; 5Departamento de Psicología Cognitiva, Social y Organizacional, Universidad de La Laguna, 38200 San Cristóbal de La Laguna, Spain

**Keywords:** approach/avoidance intentionality, relationship action-sentences, tDCS, reading, superior temporal sulcus

## Abstract

In this case, 62 university students participated in the study, in which a between-subjects design was adopted. Participants were also given the behavioral approach system (BAS) and behavioral inhibition system (BIS) scales. Participants had to read a list of 60 sentences with interpersonal and neutral content: 20 approach (“Pedro accepted Rosa in Whatsapp”), 20 avoidance (“Pedro Blocked Rosa in Whatsapp”) and 20 neutral (“Marta thought about the causes of the problem”). After reading them, they were subjected to 20 min of transcranial direct current stimulation (tDCS) in one of the two conditions: anodal (31) or sham (31). After tDCS, they had to read other list of 60 sentences matched in approach, avoidance and neutral contents with the former list. We found significant improvement in reading speed after anodal stimulation for social and neutral sentences. Regarding affective traits, we found that anodal stimulation benefitted reading speed in low-BIS and low-BAS participants and had no effect in either high BAS or high BIS participants. In addition, tDCS improvement in reading speed was significantly lower in avoidance sentences in low-BIS (avoidance) participants. We discuss these results at the light of previous research and highlight the importance of approach and avoidance traits as moderators of tDCS effects.

## 1. Introduction 

Intentionality is a basic component of understanding the minds and behaviors of others. In this regard, the temporal lobe (anterior temporal lobe, superior temporal sulcus, middle and superior temporal gyrus) and the precuneus and temporo-parietal junction constitute a “mentalizing” network [1,2,3] that encodes intentionality. It is relevant to distinguish between representation of intentions as mental states not associated with current actions, and representation of intentions and goals that are inherent in perceived actions. The latter involves a neural system particularly associated with the Superior Temporal Sulcus (STS) and is recruited for action understanding [4]. Approach and avoidance intentionality are at the serve of regulating adaptive conduct. Other pieces of research have examined relevant aspects of cognition and adaptive behavior, as its relationship with affective stimuli processing. For example, it has been suggested that medio-frontal negativity, a component of the event-related brain potential generated in ACC/mPFC, tracks the timing of salient events and reports an error signal for outcomes occurring at unexpected times [5]. Similarly, results of [6] showed how the ventromedial prefrontal cortex (vmPFC) is involved in the acquisition of fear conditioning (i.e., learning), as it is fundamental for the evaluation and representation of action’s value needed to produce sustained conditioned physiological responses. Furthermore, findings from [7] confirmed the modulation effect that emotional stimuli have on executive control, since they interfere with inhibitory control of behavior, given situational demands. Moreover, activation of this mentalizing network to process social information is usually stronger in the right hemisphere [8,9].

Previous research has highlighted how the interpretation of social signs is essential in guiding appropriate behaviors. Solid psychological evidence suggests that proximity of fear stimuli (angry faces) evoked redirection of attention. This attention redirection would be at the serve of enhancing the defensive function of the so-called Peripersonal Space (PPS) [10]. Similarly, results from a recent study [11] show that interpretation of potentially threatening situations, such as others’ proximity, triggers a number of physiological responses that help to regulate the distance between ourselves and others during social interaction. Within the mentalizing network, the STS and brain areas around it have been shown to be particularly involved in processing communicative intention for interactions by means of gaze (direct vs. averted) in social perception [12,13,14,15,16] and mutual liking [17]. It has also been shown that approach intentionality causes greater activation of posterior right Superior Temporal Sulcus (rSTS) than avoidance. In a fMRI study [18], brain activation in response to a stranger initiating or avoiding social interaction was measured. Participants viewed an animated character approaching down a virtual hallway, who shifted his gaze either toward or away from the participant. Mutual gaze (approach) caused a greater activation in this brain region than averted gaze (avoidance). These studies usually focused on demonstrating that STS is responsible for action intentionality and social contexts, and not just for the more physical aspects of actions [12,13]. However, it could be that the STS is a brain area specifically recruited for processing intentionality for relationships.

Whereas social perception of approach/avoidance intentionality activates posterior aspects of rSTS, several studies have supported that more abstract and conceptual processing of relationship intentionality recruits more anterior to middle aspects of rSTS. For example [19] (see also [20]), using a version of the Heider and Simmel animation task in a fMRI study, reported activation of more anterior aspects of rSTS when participants judged “friendship” from simple geometric shape interactions. Similarly [4] have reported activation along the full length of rSTS when participants observed Heider and Simmel animations and made social intentional judgements of interactions.

Beyond action observation, language describes how individuals interact with other people by means of social actions that conceptually involve approach (“pro-stimulus”) and avoidance (“against-stimulus”) intentionality [21,22]. For example, “Alejandro accepted/rejected Marta in his group”. Approach and avoidance would constitute a semantic frame or category to be systematically encoded for understanding this type of actions, which shows an individual’s intentional direction towards other people, and has an adaptive role. Thus, if approach/avoidance gives meaning to relationship actions, we could expect a greater activation in more anterior aspects of STS to process them in social relationship actions.

### 1.1. Non-invasive Brain Stimulation (NIBS)

Non-invasive brain stimulation (NIBS) techniques are widely used to modulate activity of brain areas. Within these techniques, transcranial electrical stimulation (tES) and transcranial magnetic stimulation (TMS) are two of the most known types of NIBS that modulate neural activity in a different way each [23]. In case of TMS, a short electromagnetic current is generated in the brain, which derive in supra-threshold activations in neurons. However, tES does not generate action potential in neurons but modulate neurons’ activity by producing sub-threshold modulations of membrane potentials [24]. In case of tDCS, direct electrical current is applied through scalp, penetrating skull and modulating cortical excitability [25]. Duration of these effects depend on parameters of stimulation, such as current density, stimulation duration and/or geometrical montage of electrodes, as well as direction of effects strongly depends on polarity [26].

Hence, selectively changing neuronal activity may be linked with modulation in cognitive functions. An example is an experiment of [27], who demonstrated that NIBS delivered on the prefrontal cortex may disrupt physiological response associated with fear memories. They run an experiment in which participants were conditioned to pictures of different indoor scenes by applying an electrical mild shock to the left inner wrist. On the next day (24 h after acquisition), reactivation of the association learned was carried out by showing the conditioned image with no electrical shock, but repetitive TMS was applied 10 min after the reactivation. In 2 out of 5 groups where reactivation occurred, rTMS was applied either on the right or left dorsolateral prefrontal cortex (r-dlPFC, l-dlPFC). On a third day (48 h after acquisition), there was a recall test where there was no association between the electrical shock and the conditioned images. For all the three phases, skin conductance response (SCR) was recorded. Statistical analysis showed that differences in SCR between signaled and non-signaled conditioned stimuli were significantly decreased during the recall test due to rTMS administration on both, right and left dlPFC, in comparison to other groups with different rTMS location, sham or recall test assessed on the second day.

However, the application of NIBS is not only on normal, but also in clinical population. For example [28], applied tDCS in 30 schizophrenic patients. Half participants received sham stimulation and the other half received tDCS at 2 mA for 20 min to see the effects of tDCS on auditory hallucinations. The electrodes were placed according to 10/20 system. Anode was placed at a point in between F3 and FP1, targeting dorsolateral prefrontal cortex, and the cathode was placed at point midway T3 and P3, targeting left temporo-parietal junction. They found a significant reduction of auditory hallucinations in the tDCS group 5 days, 1 month and even 3 months after tDCS treatment. Moreover, they found a significant reduction in negative dimension of schizophrenia symptomatology according to PANSS (Positive and Negative Syndrome Scale) was found 5 days after the treatment sessions. Other neuropsychiatric disease in which the use of NIBS has been researched is depression (see [29]. For instance [30], found an improvement in mood in depression patients in scores of the MADRS (Montgomery-Asberg Depression Rating Scale) by comparing those patients assigned to a sham-tDCS group with those patients in an actual tDCS group. They placed the anode on the left dorsolateral prefrontal cortex (which was identified as fP3) and the cathode was placed over the lateral parts of the contralateral orbit, which was localized at F8 position accordingly to the 10/20 system.

Regarding tDCS effects on the STS and relationship-actions processing, it has been found [31] found a greater improvement in discriminability of approach sentences in a memorization task after applying anodal tDCS on the rSTS, in comparison to either sham or cathodal stimulation [31]. However, the question whether the advantage of approach content could start before memorization, such as during sentence reading remains open. Likewise, in this study [31], only approach vs. avoidance sentences were tested without including neutral sentences.

In the field of reading processes, tDCS has been applied and led to significant improvement in different reading subprocesses in both normal readers and readers with dyslexia (see [32]). In these studies, a typical target area is the left temporo-parietal cortex. With regards to reading efficiency, it has been evaluated whether reading efficiency could be improved by applying tDCS on the temporo-parietal junction on both hemispheres, localized at CP5 or CP6, according to the 10–20 EEG system [33]. It was found that anodal stimulation caused a greater effect than cathodal stimulation in both reading efficiency and speed for single words.

As can be seen, effects of tDCS on reading efficiency have been examined in left temporo-parietal areas. Language is dependent on left hemisphere activity. By contrast, right hemisphere is specialized in social processing, and the temporal area around the STS is in processing social intentionality [8,9]. In accordance with the Embodied Simulation Theory [34,35], understanding social relationship action-sentences would involve experiential simulation of approach/avoidance intentionality. For that reason, we examine for the first time the improvement of reading speed in sentences with social contents by stimulating the temporal area in the right hemisphere.

### 1.2. Moderation of tDCS Effect by Approach and Avoidance Personality Traits

We also explored whether an effect of tDCS on reading speed could be moderated by approach and avoidance traits (behavioral approach system (BAS) and behavioral inhibition system (BIS) [36]. Previous research has found that only low approach trait participants show improvement following tDCS [37,38,39].

We consider that the moderator effect of trait could be exerted by affecting attention allocation in possibly two different ways. One way is by a motivational bias. In this case, we expected that greater cognitive resources furnished by tDCS would be used more on processing approach sentences than avoidance by high BAS participants, whereas high BIS participants would use more resources processing avoidance sentences. Thus, we predict greater tDCS effect on reading speed in approach than in avoidance sentences for high BAS trait, and in avoidance than in approach sentences in high BIS trait. The other way is related to a deficit in attention allocation. Previous research has suggested that high approach trait (impulsivity) is associated with less concentration, more distractibility, and less attentional narrowed focus on a given task [38,40,41]. In the case of avoidance trait, previous research has clearly showed that fearfulness and anxiety disturb the capacity for allocation of attentional resources to a particular task (see [42]). Thus, we would expect high-BAS and high-BIS traits participants to be less able to take advantage of additional processing resources plausibly furnished by anodal tDCS for the reading task. Thus, we predict a poorer reading improvement both in approach and avoidance sentences in high BAS and BIS compared to low BAS and BIS participants.

## 2. Materials and Methods

### 2.1. Design 

A 3 × 2 factorial design was used, with Direction (approach vs. avoidance vs. neutral) as within-subjects factors and Stimulation (anodal vs. sham) as between-subjects factor. The dependent variable was improvement in reading times that was measured as the difference in reading times before and after tDCS.

### 2.2. Participants 

In this case, 62 undergraduate students (54 females, *M* = 19.95, *SD* = 2.33) from the University of La Laguna (La Laguna, Spain) participated in the experiment in exchange for course credits. The minimal sample size to generate appropriate statistical power (80) with 0.05 alpha bilateral for a small effect size (*η*^2^= 0.026) for two independent means (anodal group vs. sham) was established at *N* = 31 for each group (see [43]). Inclusion criteria included: being right-handed according to the Edinburgh Handedness Inventory [44]. Exclusion criteria were suffering from epilepsy (or having close relatives affected), migraine, brain damage, cardiac, neurological or psychiatric disease, having any injury or subcutaneous metal in any of the two parts where electrodes would be set. In this study, 31 participants were subjected to the anodal condition and 31 to the sham condition.

### 2.3. Stimuli 

We selected a pool of 120 sentences from Marrero et al. [21], 80 approach and avoidance sentences that had been controlled for linguistic factors such as sentence length and number of syllables, and psycholinguistic factors such as their imaginability, and 40 neutral sentences with no social content. Table 1 showed examples of sentences in the different conditions. Among these sentences, 60% of proper names were female and 40% were male names. There is greater percentage of female students in Psychology degree. Thus we could expect more everyday female interactions which we try to translate into sentences.

### 2.4. Affective Tests

The behavioral inhibition system (BIS) and behavioral activation system (BAS) scales were measured by the Carver and White scales [45]. BAS measures individual sensitivity to reward, and BIS sensitivity to punishment [36]. Both BAS and BIS scales were reliable in this study: α = 0.851 and α = 0.825, respectively.

### 2.5. Procedure 

#### 2.5.1. Experimental Task

The approach and avoidance content of the pool of 120 sentences from Marrero et al. [21] was counterbalanced in two lists. If a sentence is approach in one list, then it appeared as avoidance in the other list, and vice versa. Then we split each list into two lists of 60 sentences each (20 sentences for each type of sentence: approach, avoidance, and neutral), one to be passed before tDCS stimulation, and the other after stimulation inasmuch we were interested in measuring reading improvement after tDCS. The order of the lists was counterbalanced as follows: list 1-tDCS-list 2; list 2-tDCS-list 1; list 3-tDCS-list 4; list 4-tDCS-list 3. Participants were randomly assigned to each sequence order, taking into account that they were all from the first course of Psychology degree, and thus are assumed to be homogeneous in cognitive reading skills. Sentences were randomly presented to the participants in each of the counterbalanced sets. Participants were told that the task consisted of reading segmented sentences one by one displayed on a computer screen for comprehension, while they were seating in front of it.

At the start of the experiment, participants were given seven sentences to practice. Then, they were given 60 sentences, 20 for each experimental within-subject condition: Approach vs. Avoidance vs. Neutral. Each sentence presentation started with a cross point displayed in the middle of the screen for 750 ms. After an interval of 150 ms, one sentence was displayed. Sentences presentation was segmented (three segments, see Table 1); for example, “Pedro/bloqueó a Rosa/en el Whatsapp” (“Pedro/blocked Rosa/in Whatsapp.”). Each segment was displayed till the participant pressed the corresponding button. After 750 ms a new sentence appeared. To avoid participants’ superficial reading, 36 sentences were immediately followed by a question on the contents just read (e.g., “Is it stated that Pedro blocked Rosa?”). This question had either a positive or a negative response half the times and remained on the screen for 5000 ms or until a response was made. Feedback on correctness and time required was given to the participants and displayed for 2000 ms These questions were aimed at keeping the attention of participants on reading comprehension. After a delay of 750 ms, a new sentence was displayed. Response recordings and stimuli presentation were controlled by E-Prime 2.0 software (Psychology Software Tools, Pittsburgh, PA, USA).

#### 2.5.2. Protocol for tDCS Application

A CE-certified battery-powered stimulator (neuroConn DCSTIMULATOR. neuroConn GmbH, Albert-Einstein-Str. 3, 98693 Ilmenau, Germany) was used for the non-invasive tDCS current conduction with an intensity of 2 mA. The electrodes of the equipment used were rubber, with one being 5 × 5 cm and 7 × 5 cm the other. Both were covered with sponges soaked in saline to transfer direct current, which resulted in a density of 0.08 mA/cm^2^ and 0.057 mA/cm^2^, respectively. The smaller electrode was placed on the scalp in accordance with International System 10–20. The selected area was T8, as it is the most appropriate for the stimulation of the temporal region of interest. The other electrode was extracranially placed on the contralateral shoulder to minimize its effects on the brain and have higher focality at the region of interest [46]. We stimulated BA 22 and BA 21 brain areas overlapping medial aspects of rSTS, as shown in Figure 1. In addition, the stimulated area is a part of the so-called mentalizing network [2], specialized in processing social intentionality.

The stimulation application time was 20 min plus a Fade in and Fade out of 15 s both. The stimulation time was established based on previous studies of tDCS (e.g., [48,49]). During the false tDCS (sham) condition, the constant current only lasted 45 s: Fade in: 15 s, 15 s maximum intensity and Fade out: 15 s.

#### 2.5.3. tDCS Procedure

Upon arrival at the laboratory, participants were informed about the general aim of the study. They all filled in a personal data form and a questionnaire to screen for exclusion conditions and signed an informed consent form. Participants were told that the objective of the study was to examine the effect of brain stimulation on cognitive enhancement. They were not informed about the tDCS condition they had been assigned to. Thus, in both tDCS conditions, they were supposed to believe that they were being positively stimulated. No direct assessment of blinding assessment was performed. None of them reported suffering from epilepsy (nor having close relatives affected), migraines, brain damage, cardiac disease, or other psychological or medical conditions. All participants were right-handed, according to the Edinburgh Handedness Inventory [44]. The ethical committee of the University of La Laguna approved the study: (CEIBA 2017–0272).

Participants were also given the BIS/BAS scales. Subsequently, they performed the first set of sentences of the experimental task. After that, electrodes were placed, and tDCS stimulation started in accordance with the tDCS protocol. Immediately after removing the tDCS equipment, participants performed the second set of sentences. Once this task was finished, they were thanked for their cooperation, and a short explanation of the experimental procedure was given to them for debriefing. Likewise, they were advised not to discuss the experiment with other potential participants.

The experimental session lasted approximately 55 min. The stimulation parameters were considered safe [50]. We asked participants to inform us of any adverse event during tDCS application. We asked the subjects again at the end of the experimental session and told them to let us know whether they felt such effects in the following days. Some volunteers informed us of mild and transient adverse effects (see [51]) during intervention. Table 2 shows the type of adverse effect, the severity of the effect and the percentage of the participants that experienced them.

## 3. Results 

Sentence reading times above/under 2.5 SD of the participant mean (1.8%) were removed from the analysis. Two participants were removed from the analysis as they exceeded the criteria of less than 25% of incorrect responses to the questions.

We assumed a normal distribution of improvement in reading speed. The Saphiro-Wilk test supported a normal distribution of improvement (*p* > 0.05). We carried out an ANOVA with Stimulation (anodal vs. sham) as a between-subjects factor and Direction (approach, avoidance and neutral) as a within-subjects factor. We used the latency to question in neutral sentences after tDCS as a covariate to further control attentional variability in the reading task. Descriptive data of reading improvement are shown in Table 3. Likewise, in Figure 2, the score distributions for tDCS conditions in each type of sentence are shown.

The main effect of Stimulation was significant, F(1,58) = 4.174, *p* < 0.046, *ƞp^2^* = 0.068). Anodal stimulation improved reading speed for all the types of sentences in contrast to sham condition (see Table 3). The main effect of Direction was marginally significant, F (2,59) = 2.896, *p* = 0.064, *ƞp^2^* = 0.094. Reading improvement was greater for approach than for avoidance sentences (M_Diff_. = 129.521, *SD* = 225.479), *t*(57) = 4.45, *p* < 0.001; improvement was also greater for approach than for neutral sentences *(M*_Diff_. = 76.823, *SD* = 318.953 ), marginally significant, *t*(57) = 1.866, *p* = 0.087; and for neutral than for avoidance sentences (*M*_Diff_. = 52.697, *SD* = 223.675), marginally significant, *t*(57) = 1.82, *p* = 0.073. The interaction Direction × Stimulation was not significant, *p* > 0.10.

### 3.1. Moderation of tDCS by Affective Traits

We examined modulation by affective traits of tDCS effect on reading improvement. Modulatory analyses are aimed at examining whether tDCS affected reading performance of participants depending on having a higher or lower levels either in BAS or in BIS traits. Participants were classified in BAS trait as ‘low’ (those who scored below the 35th percentile score of 1.70), ‘medium’ (between the 35th and 65th percentile), and ‘high’ (higher than the 65th percentile score of 2.14) taking into account the whole sample, and the range was 1.15–3.38. Likewise, they were classified in BIS trait as ‘low’ (those who scored below the 35th percentile score of 2.01), ‘medium’ (between the 35th and 65th percentile), and ‘high’ (higher than the 65th percentile score of 2.42) taking into account the whole sample, and the range was 1.43–3.29. We were interested in looking for differences between the low-high trait participants, therefore, intermediate levels of each trait were not of interest.

### 3.2. Behavioral Approach System (BAS)

Saphiro-Wilk test supported a normal distribution of d scores in participants for both Stimulation conditions (*p* > 0.05) and the two BAS groups. Following the same design as described above, we carried out two 2 × 3 ANOVAs on reading improvement. In the case of low-BAS participants, a main effect of Stimulation was found, F(1, 19) = 6.53, *p* = 0.02, *ƞp^2^* = 0.205. As can be seen in Table 4, anodal stimulation furnished greater improvement than sham condition in the three types of sentences. Main effect of Direction and the interaction Direction × Stimulation were not significant (*p* > 0.5).

By contrast, no main effect of Direction, Stimulation or the interaction Direction × Stimulation was found in the case of high-BAS participants (Anodal: 11; Sham: 10), *p* > 0.10.

### 3.3. Behavioral Inhibition System (BIS)

The same procedure as for the BAS trait was applied. Saphiro-Wilk test supported a normal distribution of reading improvement scores in participants for both Stimulation-conditions (*p* > 0.05) and the two BIS groups. In the case of low-BIS participants, a main effect of Stimulation was found, F(1,19) = 8.502, *p* = 0.009, *ƞp^2^* = 0.321. As can be seen in Table 5, anodal stimulation furnished a greater improvement than sham condition in the three types of sentences. Likewise, the interaction Direction × Stimulation resulted marginally significant, F(2,19) = 3.181, *p* = 0.067, *ƞp^2^* = 0.272. Follow-up comparisons showed that in anodal condition reading improvement was less in avoidance sentences than in approach sentences, *t*(10) = 2.92, *p*= 0.015; and less in avoidance sentences than in neutral sentences, *t*(10) = 2.575, *p* = 0.029. By contrast, there were no significant differences in the sham condition. The main effect of Direction was not significant, *p* = 0.174.

In the case of High-BIS participants (Anodal: 11, Sham: 11), the main effect of Direction resulted marginally significant: F(2,21) = 3.077, *p* = 0.071, *ƞp^2^* = 0.255. Reading improvement tended to be greater in approach sentences (*M* = 265,03, *SD* =256.19) than in avoidance sentences (*M* = 202.61, *SD* = 247.538) or in neutral sentences (*M* = 240.10, *SD* = 311.96). Neither the main effect of Stimulation nor the interaction Direction × Stimulation resulted significant.

## 4. Discussion 

We examined the effect of anodal tDCS on reading improvement of social relationship sentences, both of approach and avoidance, and those without interpersonal content (neutral sentences). We found an effect of training with greater improvement for approach than for avoidance or neutral sentences. This effect was not affected by tDCS. Thus, this effect plausibly shows a certain facility to process approach and so, faster reading improvement in approach sentences from training in contrast to avoidance ones. In this regard, previous research [21] have found that avoidance sentences are judged as emotionally negative and more arousing than approach ones. In the case of words, more negative emotionality (see [52]) has been associated with a greater effort in reading. Importantly, we found a main effect of tDCS. Anodal stimulation improved reading speed in the three types of sentences in contrast to sham condition: as mentioned, tDCS effect was not modulated by the type of sentence.

The aim of our study has been to examine whether the advantage of approach content in the tDCS effect on sentence memorization [31] starts before, during sentence reading. Contrary to our expectations, reading speed of approach sentences did not benefit from anodal stimulation in rSTS. One plausible explanation is that the positive effect of anodal stimulation on memorization of approach content occurs after reading, during the process of encoding the meaning of the sentences. In this case, anodal stimulation would benefit cognitive accessibility of approach for better memorization. This hypothesis could be examined by using an immediate recognition task. Recognition tasks are suitable to examine content accessibility after reading (see [53]). Further research is thus necessary to study whether anodal tDCS on rSTS benefits memorization of approach sentences by enhancing the availability of approach content after reading.

As mentioned, anodal stimulation in rSTS had a general effect on reading improvement. That is, tDCS seems to furnish cognitive resources to reading for the three types of content: interpersonal (approach and avoidance) and neutral content. One possible explanation is that stimulation facilitated semantic processing in general, and it does not depend on the content. However, we must consider that we stimulated the right temporal area that is specialized in processing social content. Thus, we consider plausible that the three types of sentences share social content, as neutral sentences although they are not interpersonal, referring to persons. In this regard [9], have shown that more anterior and superior aspects of the right temporal area is recruited for sentence processing and person content. Thus, stimulation of medial to anterior aspects of rSTS would have an effect on reading the three types of sentences.

### 4.1. tDCS Effect on Reading Speed Improvement Is Modulated by Approach/Avoidance Trait

In terms of Approach-BAS modulated tDCS effect on reading speed, low-BAS participants showed a significant effect of anodal stimulation in reading improvement in comparison to Sham condition participants. By contrast, high BAS participants did not show any effect of tDCS on reading improvement. This result is in accordance with previous research that found a greater effect of anodal stimulation in low-approach (BAS) participants [37] and supports the attentional explanation, although rules out the motivational one. High approach (reward sensitivity) has been associated with less concentration, more distractibility, and less attentional narrowed focus on a given task [38,40,41]. Thus, high BAS participants would be less able to take advantage of additional processing resources plausibly furnished by anodal tDCS to read sentences, compared to low-approach ones.

Avoidance (BIS) trait also modulated the tDCS effect. Low-BIS participants showed a significant effect of anodal stimulation in reading improvement in comparison to sham participants. By contrast, high-BIS participants showed no effect of tDCS on reading improvement. This result also supports our attentional hypothesis. Fearfulness and anxiety would disturb the capacity for allocation of additional processing resources furnished by anodal stimulation to the task (see [40]).

Interestingly, we found that the effect of tDCS on reading improvement in low-BIS participants was modulated by the type of sentence. Post-hoc comparisons showed that anodal tDCS is associated with lesser improvement in avoidance sentences. This suggests a motivational bias but in the opposite direction of our motivational hypothesis. That is, participant with low-BIS (fear and anxiety) seem to be less benefited in reading speed of avoidance sentence from anodal stimulation. One plausible reason is a motivational bias: low-BIS participants paid less attention to avoidance content, and so took less advantage of cognitive resources furnished by anodal stimulation to increase their reading speed in the task.

### 4.2. Limitations

A limitation of our study is the lack of focality of the applied stimulation. For the anatomical localization of the STS, we considered the position of electrode T8 of the EEG montage; however, aspects such as the anatomical variability across subjects and the lack of focality of the applied stimulation could have played an important role in the results. Moreover, our participants were university students with a high percentage of females, and they were all young participants. However, approach and avoidance brain processing could be affected by developmental changes or modulated by gender. Thus, future studies should also include adult and more male participants.

## 5. Conclusions

Anodal stimulation on rSTS had no effect on reading approach content; however, it did have a general effect on reading social content about persons. Plausibly, the positive effect of anodal stimulation on memorization of approach occurs after reading, during the process of encoding the meaning of the sentences. BIS and BAS affective traits modulate tDCS effect on reading speed, which benefits low BAS and BIS participants, plausibly due to problems with allocation of additional resources in high BAS and BIS participants. Likewise, low-BIS participants seem to show a motivational bias towards paying less attention to avoidance content. In short, anodal tDCS appears to improve reading speed of social content, but not specifically of interpersonal or approach content. The modulation of affective traits in modulating tDCS effects has emerged as a relevant factor. More attention to this modulation would be necessary in future research on cognitive improvement by tDCS stimulation. Our results are relevant for brain research on the mentalizing network to understand social content in sentences and its relationship with differences in personality traits.

## Figures and Tables

**Figure 1 brainsci-11-01464-f001:**
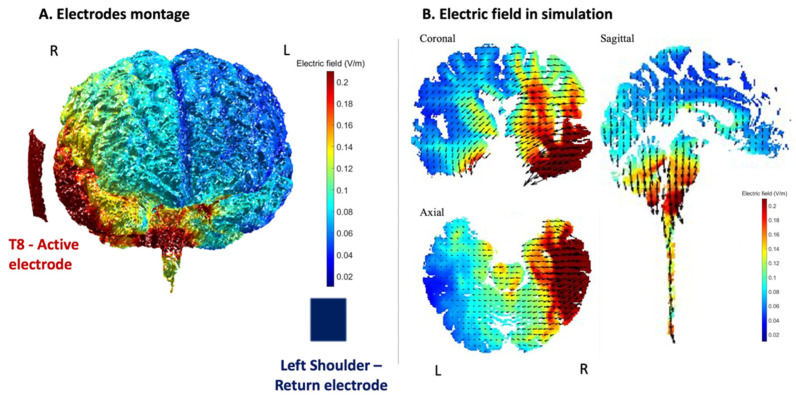
Computational representation of the electric field intensity generated by our transcranial direct current stimulation (tDCS) montage with reference to anode (T8) and an extracephalic cathode. Units are in V/m. The simulation was run using ROAST (realistic volumetric approach to simulate transcranial electric stimulation) [47]. (**A**) Electrode montage and, (**B**) Electric field in simulation in three slides (axial, coronal and sagittal planes).

**Figure 2 brainsci-11-01464-f002:**
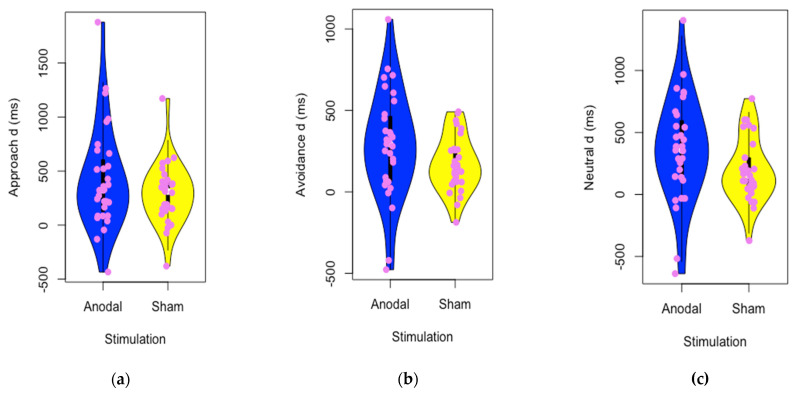
Distribution of speed reading improvement for tDCS conditions in the three types of sentences: approach (**a**), avoidance (**b**) and neutral (**c**).

**Table 1 brainsci-11-01464-t001:** Examples of approach, avoidance and neutral sentences with questions.

Sentence	Direction	Question Example	Correct Answer
Pedro/aceptó a Rosa/en Whatshapp (Pedro/accepted Rosa/in Whatsapp)	Approach	¿Dice que Pedro aceptó a Rosa en Whatshapp? (Is it stated that Pedro accepted Rosa in Whatshapp?)	Yes
Pedro/bloqueó a Rosa/en Whatshapp (Pedro/blocked Rosa/in Whatshapp	Avoidance	¿Dice que Pedro aceptó a Rosa en Whatshapp? (Is it stated that Pedro accepted Rosa in Whatshapp?)	No
Verónica/dedujo el precio/del abrigo (Verónica/deduced the price/of the coat)	Neutral	¿Dice que Verónica dedujo el precio del abrigo? (Is it stated that Verónica deduced the price of the coat?)	Yes

**Table 2 brainsci-11-01464-t002:** Adverse effects, severity, and percentage of participants that experienced them in the tDCS study.

Type of Effect	Severity	Percentage
Tingling	Mild	27.14%
Itching	Mild	67.14%
Warm	Mild	4.28%

**Table 3 brainsci-11-01464-t003:** Descriptive statistics of reading improvement as a function of the type of sentence and the tDCS conditions.

Direction	Stimulation	Mean	*SD*	*N*
Approach	Anodal	425.81	465.06	31
	Sham	288.82	280.42	29
Avoid.	Anodal	284.83	327.73	31
	Sham	171.56	170.30	29
Neutral	Anodal	363.5	418.53	31
	Sham	196.49	255.64	29

**Table 4 brainsci-11-01464-t004:** Descriptive statistics of d for low-BAS participants in each condition.

Direction	Stimulation	Mean	*SD*	*N*
Approach	Anodal	490.37	412.64	11
	Sham	206.099	309.16	10
Avoid.	Anodal	334.41	217.85	11
	Sham	87.13	166.84	10
Neutral	Anodal	411.12	420.50	11
	Sham	81.03	228.46	10

**Table 5 brainsci-11-01464-t005:** Descriptive statistics of d for low-BIS participants in each condition.

Direction	Stimulation	Mean	*SD*	*N*
Approach	Anodal	583.70	412.52	11
	Sham	280.76	417.18	10
Avoid.	Anodal	366.82	263.08	11
	Sham	169.39	188.28	10
Neutral	Anodal	562.93	418.99	11
	Sham	130.92	280.96	10

## Data Availability

The data presented in this study are openly available in Figshare at 10.6084/m9.figshare.16867144.

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
