# Peer review of "Personality Traits Modulate the Effect of tDCS on Reading Speed of Social Sentences"

_brainsci, 2021, doi:10.3390/brainsci11111464_

Round 1

Reviewer 1 Report

Overall, I find the paper interesting, and I believe it tackles some unexplored questions. However, I have several (mostly methodological) comments I believe authors need to better address in the paper for it to be a significant addition to the current tDCS literature.

  • I complement the authors for doing the power analysis – this is something we really need in all tDCS studies. However, the effect size for which the power was calculated was not reported. Also, it is unclear if the N of 44 is total or per group. If it was total – why did you go for 60+. My experience is that with typical tDCS you cannot have good power in between-subject design with 22 participants per group. But since the size of the effect is not presented here, I cannot judge if the power calculation was done correctly or not (doesn’t seem correct, but I might be wrong so please provide all information, so your power analysis is replicable)
  • How did you randomize participants across groups? Was it random allocation or were they matched – in any case, please explicitly state your rationale behind the choice and explain if and how that might affect your results.
  • Is there any rationale behind this: 60% of proper names were female and 40% were male names?
  • I believe the manuscript would be easier to follow if the 2.3. Design would be the moved at the beginning of the method (e.g. To understand power analysis, one needs to know the design).
  • I am not sure I understand the experimental task. Authors state that there is total of 120 sentences; than they state that there are 4 sets of sentences - is it split so it is 30 sentences per participant? How come that we then have 60 sentences per participant? Was the order of stimuli presentation random for enc participant or rerandomized for each set – i.e. did all participants who had same set see sentences in the same or different order? Why was feedback given (Feedback on correctness and time required was given to the participants and displayed for 2000 ms)? Please provide a rationale for giving feedback especially on the response time. And finally – most important I don’t understand what we're participants completing before and what after tDCS? The author just state it was a “similar task “. This part needs to be clarified as it is central to the study.
  • It is not often to use different sizes of electrodes for anode and return – please provide a rational for this decision (I agree it is a good way to decrease the cutaneous sensations at the irrelevant area, and have higher focality at the region of interest, or is the result of the modeling). Whatever it is, it is important to provide rationale in the manuscript.
  • Who did you do this modeling? As far as I am aware COMETS (2013) cannot model the field with extracranial return electrodes (it would need a whole-body mash for that). Is it some specific add-in you used? Please provide additional modeling images – both left and right side and the top and bottom view. At least one of the pictures should have electrodes as well. I just want to seed that the return electrode is not on the other side of the brain ?
  • Provide the number of IRB approval at the end of this sentence: The ethical committee of the University of La Laguna approved the study.
  • TDCS blinding is another important issue – how was blinding assured and how was the blinding assessed?
  • How were participants split into 3 groups depending on the BIS/BAS? What are the 3 groups? What were the cut of scores? What are the 2 BAS groups then? How come 62 participants were divided – how was the division in respect to sham and active group?

Specific or minor comments:

  • As abstract is the gateway for your paper, I believe this part should be rewritten as it might seem like contradictory findings (when someone is not fully aware of your design): We found significant improvement in reading speed after anodal stimulation for social and neutral sentences; Against our expectations, anodal stimulation of right temporal area did not specifically improve reading speed of approach sentence I would suggest clearly stating your conditions earlier in the abstract.
  • STS as abbreviation is introduced twice (page 1)
  • (see Cancer and Antonietti [19]) should simply be [19]
  • I have difficulties understanding the sentence “Left hemisphere, particularly Wernicke area, is linguistic”. You clearly wanted to state that language is dependent on left hemisphere activity – please rephrase it
  • You state in the introduction “Although our stimuli are sentences, the con-tents are social, and readers would process social intentionality by experiential simulation in accordance with EST”. However, the reader still does not know your methodology because it is described later. To increase communicability of your paper, rephrase this part so it does not make reader scroll back and forth.
  • “Previous research has found” – it should be “have”
  • This is unclear: “low approach trait benefitting from anodal stimulation”; do you mean the effect of tDCS is moderated by some trait? that the tDCS effect size correlates with the score on the BAS/BIS? That only those with low BAS scores show improvement following tDCS?
  • Carver & White (32) – should be properly cited.

I would strongly encourage authors to  improve the methodological aspect of the paper - because without those clarifications I  cannot recommend it for publishing 

Reviewer 2 Report

In the present study entitled ‘The effect of tDCS on the rSTS on reading speed of social sentences is modulated by personality traits’, by Reyes and colleagues, authors aimed to investigate the effect of transcranial direct current stimulation (tDCS) on the right superior temporal sulcus (rSTS) on improving reading speed for relationship-action sentences. For this purpose, 62 participants had to read sixty sentences (20 approach, 20 avoidance and 20 neutral) and then received either anodal or sham tDCS stimulation, based on group allocation. Results showed that anodal stimulation improved reading speed for social and neutral sentences, but not for approach sentences.

In general, I think the idea of this research article is really interesting and the authors’ fascinating observations may be of interest to the readers of Brain Sciences. However, some comments, as well as some crucial citations that should be included to support the authors’ argumentation, need to be addressed to improve the article, its adequacy, and its readability prior to the publication in the present form.

Comments

  • I suggest changing the title. In my opinion, in the present form, it seems to be too wordy. An example could be “Personality traits modulate the effect of tDCS on reading speed of social sentences”.
  • According to the Journal’s guidelines, authors should have provided an abstract of about 200 words maximum. Indeed, the current one includes 222 words. Furthermore, the keywords are missing.
  • Page 1, Introduction: I think that the statement “It is relevant to distinguish between representation of intentions as mental states not associated with current actions and representation of intentions and goals that are inherent in perceived actions” needs some citations. In this regard, I would suggest adding some critical studies that examined the role of different brain areas in the interpretation of stimulus-outcome contingencies: decisive evidence from Garofalo and colleagues’ study (2017, Journal of Cognitive Neuroscience) suggests that medio-frontal negativity, a component of the event-related brain potential generated in ACC/mPFC, tracks the timing of salient events and reports an error signal for outcomes occurring at unexpected times. Similarly, one of the latest studies by Battaglia and colleagues (2020, The Journal of Neuroscience) show how the ventromedial prefrontal cortex (vmPFC) is involved in the acquisition of fear conditioning (i.e., learning), as it is fundamental for the evaluation and representation of action’s value needed to produce sustained conditioned physiological responses. Furthemore, findings from Battaglia and colleagues’ review (2021, Behaviour Research and Therapy) confirmed the modulation effect that emotional stimuli have on executive control, since they interfere with inhibitory control of behaviour, given situational demands.
  • Page 1-2, Introduction: The introduction is well written and organized; however, in my opinion, a general overview about non-invasive brain stimulation (NIBS) widely used to modulate brain activity and how they are applied to alter various cognitive domains is needed to give non-expert readers an adequate background about the topic. Thus, I would suggest some references that would be crucial and methodologically fit with the present manuscript: for example, in a recent review, Borgomaneri and colleagues (2020, Cortex) outlined how NIBS techniques can be used to manipulate action inhibition in healthy individuals. Also, I would suggest to cite results from Borgomaneri and colleagues’ recent study (2020, Current Biology) that show causal evidence for the application of rTMS over DLPFC after memory reactivation in reducing responding to learned fear. The same research group, in a recent yet critical review (Borgomaneri et al., 2021, Neuroscience and Biobehavioral Reviews) addressed the implementation of NIBS to modulate fear memories. Finally, I would suggest Borgomaneri and colleagues’ study (2021, Journal of Affective Disorders), that illustrated the therapeutic potential of NIBS as a valid alternative for those patients not responding to psychotherapy and/or drug treatments. I believe that adding information regarding different applications of NIBS will dramatically improve this section, thus I recommend it.
  • Page 2, Introduction: Authors described how social perception of approach/avoidance has an adaptive role because it is processed for understanding how people interact and individual’s intentions towards others. Still, I think that this section would benefit from some citations here that highlight how the interpretation of social signs is essential in guiding appropriate behaviors: solid psychophysiological evidence from Ellena and colleagues’ study (2020, Experimental Brain Research) suggest that responses to approaching emotional stimuli provide a cue to the presence of a threat in the environment and modulate autonomic arousal as a function of the distance between the observer and the stimuli. Similarly, results from a recent study by Candini and colleagues (2021, Scientific Reports) show that interpretation of potentially threatening situations, such as others’ proximity, triggers a number of physiological responses that help to regulate the distance between ourselves and others during social interaction.
  • Page 8, BIS results: please indicate all the p values after statistical information.
  • Regarding the figures: I suggest adding a figure that displays stimulation sites’ position on the scalp. Also, I would like to see a higher quality image of the electric field distribution in Figure 2, therefore I recommend to use the Realistic vOlumetric-Approach to Simulate Transcranial electric stimulation (ROAST) to segment the full head from an MRI structural image, place virtual electrodes and solve for voltage field distribution at 1 mm resolution. In my opinion, this would dramatically help the reader to better comprehend tDCS montage and stimulation.
  • According to the Journal’s guidelines, the authors should have included funding information, the ‘Acknowledgments’, ‘Author Contributions’, ‘Institutional Review Board Statement’, ‘Informed Consent Statement’ and ‘Conflicts of Interest’ sections and should have provided a link to the data set, where the experimental data are deposited, in order to ensure the replicability of the study.
  • The reference list is incorrect: authors should check the Journal’s guidelines again and provide only the abbreviated journal name in italics, the year of publication in bold, the volume number in italics and write the page range.
  • Overall, I advise the authors to submit their work to an English proofreader to help with some grammar and lexical errors that can be found in different sections of the manuscript.

Round 2

Reviewer 1 Report

NIce work! Thank you for addressing all my concerns. Regrading the blinding I would only state "no direct assessment of blinding success was performed".

Author Response

Thank for your comment, we have changed the paragraph as "Thus, in both tDCS conditions, they were supposed to believe that they were being positively stimulated. No direct assessment of blinding assessment was performed.".